# In Vitro Functional Characterization of Type-I Taste Bud Cells as Monocytes/Macrophages-like Which Secrete Proinflammatory Cytokines

**DOI:** 10.3390/ijms241210325

**Published:** 2023-06-19

**Authors:** Aziz Hichami, Hamza Saidi, Amira Sayed Khan, Pernelle Degbeni, Naim Akhtar Khan

**Affiliations:** 1Physiologie de la Nutrition & Toxicologie, UMR INSERM U1231 Lipide, Nutrition & Cancer, Université de Bourgogne, 21000 Dijon, France; saidi.hamza.dz@gmail.com (H.S.); amira.khan@u-bourgogne.fr (A.S.K.); pernelle_degbegni@etu.u-bourgogne.fr (P.D.); 2Bioenergetics and Intermediary Metabolism Team, Laboratory of Biology and Organisms Physiology, University of Sciences and Technology Houari Boumediene, Algiers 16111, Algeria

**Keywords:** GLAST, taste bud cells, F4/80, inflammation, type I cells, obesity

## Abstract

The sense of taste determines the choice of nutrients and food intake and, consequently, influences feeding behaviors. The taste papillae are primarily composed of three types of taste bud cells (TBC), i.e., type I, type II, and type III. The type I TBC, expressing GLAST (glutamate-–aspartate transporter), have been termed as glial-like cells. We hypothesized that these cells could play a role in taste bud immunity as glial cells do in the brain. We purified type I TBC, expressing F4/80, a specific marker of macrophages, from mouse fungiform taste papillae. The purified cells also express CD11b, CD11c, and CD64, generally expressed by glial cells and macrophages. We further assessed whether mouse type I TBC can be polarized toward M1 or M2 macrophages in inflammatory states like lipopolysaccharide (LPS)-triggered inflammation or obesity, known to be associated with low-grade inflammation. Indeed, LPS-treatment and obesity state increased TNFα, IL-1β, and IL-6 expression, both at mRNA and protein levels, in type I TBC. Conversely, purified type I TBC treated with IL-4 showed a significant increase in arginase 1 and IL-4. These findings provide evidence that type I gustatory cells share many features with macrophages and may be involved in oral inflammation.

## 1. Introduction

Taste perception of relevant food components on the tongue is mediated by specialized taste receptors expressed in sensory cells in taste buds of lingual papillae [1]. There are three types of gustatory papillae: fungiform papillae, found in the anterior part of the tongue; circumvallate papillae, localized in the central region of posterior tongue; and foliate papillae, positioned in the lateral region of the tongue [2,3]. Four types of cells compose each taste bud. Type I cells, glial-like supporting cells sharing many features with astrocytes, are involved in the perception of salty taste [4]. Type II cells, also called “receptor cells”, detect sweet, bitter, and umami taste qualities [5]. Type III cells are neuron-like cells, required for sour taste perception [6]. Finally, type IV cells are precursor cells responsible for the renewal of taste bud cells (TBC) [7,8]. Interactions between a tastant and its specific detection system generate an increase in free intracellular calcium concentration, [Ca^2+^]i, in TBC, triggering the release of neurotransmitters towards afferent nerve fibers that convey taste information to the brain via connections between the gustatory nerves and the nucleus of solitary tract in the brain stem [1,9,10].

Several studies have shown that taste perception can be altered both in obese rodents [11] and human beings [12]. Indeed, Ozdener et al. [11] showed that obese mice had a lower orosensory sensitivity for a dietary fatty acid compared to control mice in long-term double choice tests. This attenuated orosensory detection has also been demonstrated in humans by Stewart et al. [13], who reported that the detection of oleic acid within the oral epithelium is reduced in obese subjects and associated with increased high fat diet consumption. It is well known that obesity is characterized by chronic low-grade inflammation [14]. In addition, taste cells have been shown to be able to naturally express elevated levels of several inflammatory cytokines and their receptors. The inflammation induced by intraperitoneal injection of lipopolysaccharide (LPS) increases the expression of TNFα, IFN-γ, and IL-6 in circumvallate and foliate papillae. These pro-inflammatory cytokines can reduce the rate of taste bud renewal, influence the proliferation and viability of type II cells, and, therefore, alter taste perception [15].

Despite the fact that type I cells represent half of the mammalian taste bud cells, few details are known regarding their function. The present study was designed to demonstrate that type I cells can be differentiated under in vitro conditions into pro-inflammatory phenotype and produce some inflammatory cytokines. Hence, we first isolated and purified type I cells, then we characterized them under pro-inflammatory and anti-inflammatory conditions, in order to investigate if they could express M1 or M2 macrophage markers.

## 2. Results

### 2.1. GLAST-Positive Cells Express F4/80

Type I GLAST1 positive cells were purified using the anti-GLAST kit (ACSA-1) and MicroBeads. ACSA-1 antibody was generated by immunization of GLAST1 knockout mice, and it specifically detects an extracellular epitope of astrocyte-specific L-glutamate/L-aspartate transporter GLAST1 (EAAT1, Slc1a3).

Purified cells were cultured for 12 h on a coverslip and immunostained to investigate their purity and the expression of macrophage markers. Double labelling using anti-GLAST (red) and anti-F4/80 (green) antibodies showed that all GLAST-positive cells are glial-like cells that do co-express a marker of macrophages (Figure 1A). GLAST-positive cells also express NTPDase2, a marker of type I cells (Figure 1B), and keratin 8 (also known as cytokeratin 8), a general taste cell marker (Figure 1C).

Interestingly, there is no expression of GLAST in purified GLAST-negative cells which belong to taste bud cells, as they express cytokeratin 8 (Figure 1D). These observations confirm that GLAST-positive cells belong to type I gustatory cells and these cells may have immune function in taste buds.

### 2.2. GLAST-Positive Gustatory Cells Share Similarities with Macrophages

To confirm the results obtained by immunocytochemistry and to investigate the co- expression of other macrophage markers by type 1 TBC, flow cytometry analyses were performed on isolated and labeled taste bud cells.

Figure 2A shows that between 50% and 80% of gustatory cells expressed GLAST. We also observed that 30.3% of gustatory cells expressed F4/80 (Figure 2B). Furthermore, a double-labeling with anti-GLAST and anti-F4/80 antibodies revealed that 22.6% of the cells expressed GLAST and F4/80 simultaneously (Figure 2C). However, a double-labeling with anti-GLAST/anti-CD64 and anti-GLAST/anti-CD11b antibodies showed that 1.87% co-expressed GLAST and CD64, while 3.24% of the cells co-expressed GLAST and CD11b (Figure 2D,E).

### 2.3. GLAST-Positive Gustatory Cells Express mRNA of Macrophage/Glial Cell Markers

We investigated mRNA expression of additional macrophage markers by RT-qPCR in GLAST-positive cells, using GLAST-negative cells as control. As expected, Figure 3A shows that the GLAST mRNA expression was significantly higher in GLAST-positive cells compared to GLAST-negative cells. We also observed that GLAST-positive cells expressed higher mRNA encoding GLAST, CD68, CD11c, F4/80, and TLR4, compared to GLAST-negative cells (Figure 3A).

### 2.4. GLAST-Positive Gustatory Cells Can Be Differentiated under In Vitro Conditions into a Phenotype Similar to M1 or M2 Macrophages

As type 1 glial-like cells share several markers with macrophages, we wanted to determine if they can be differentiated into M1 and M2 macrophages, by measuring mRNA expression related to these phenotypes.

Treatment of GLAST-positive cells with IL-4 (polarization to M2-like) triggered F4/80 IL-4 and arginase 1 mRNA expression, curtailed TNFα expression, but did not affect IL-6 and IL-1β mRNA expression when compared to untreated cells, suggesting that GLAST-positive cells can acquire a M2-like phenotype following IL-4 treatment (Figure 3B). Conversely, treatment of GLAST-positive cells with LPS (polarization to M1-like) induced the expression of IL-1β, IL-6, and TNFα mRNA expression, the proinflammatory markers characteristics of the M1 macrophage phenotype, when compared to untreated cells (Figure 3C).

Determination of IL-4 and IL-6 concentrations in the supernatant of GLAST-positive cells (Figure 4) shows that LPS induced the secretion of IL-6 (Figure 4A) and TNFα (Figure 4B); however, IL-4 significantly decreased the secretion of TNFα compared to control (Figure 4B).

### 2.5. GLAST-Positive Gustatory Cells Are Differentiated into M1-Macrophages in Obesity

After 12 weeks of regimen, HFD-fed animals became obese with a significant increase in body weight (40.12 g ± 1.36) compared to STD-fed animals (29.33 g ± 1.22) (Figure 5A). After sacrifice, tongues were removed, GLAST-positive gustatory cells were isolated as described here-above, and mRNA expression of F4/80, TNFα, IL-6, IL-1β, TLR4, and CD68 was investigated.

RT-qPCR analysis revealed that the mRNA expression of macrophage markers F4/80, TLR4, and CD68 (Figure 5B) was significantly higher in HFD-fed animals in comparison to STD-fed animals. In addition, the mRNA expression of pro-inflammatory cytokines TNFα, IL-6, and IL-1β (Figure 5B) was significantly higher in HFD-fed animals in comparison to STD-fed animals, suggesting that type I gustatory cells can be differentiated into M1-like phenotype of macrophages in obesity.

## 3. Discussion

It has been reported that LPS-induced inflammation inhibits proliferation and shortens the life span of TBC [15]. TBC express inflammatory cytokines like IFN-γ and TNFα. Indeed, Kim et al. [16] have reported a high expression of IFN-γ in SNAP-25-positive cells (a marker of type III gustatory cells) and of TNFα in a subset of PLC-β2-positive cells (a marker of type II gustatory cells) in MRL/lpr mice which develop an autoimmune disease. As type 1 TBC are glial-like cells, it is likely that they can be differentiated like brain glial cells, according to the mediators and cytokines present in their environment [17]. Microglia activation is classified as M1 or alternative M2 [18], following the concept used for macrophages [19], where the M1 phenotype represents a pro-inflammatory and neurotoxic state, while the M2 phenotype represents an anti-inflammatory and healing state of microglia [20].

The aim of the current study was to determine if type I TBC express macrophage markers and can differentiate into M1- or M2-like macrophages. Here, we report, for the first time, that type I gustatory cells purified with GLAST antibodies express F4/80, one of the most specific cell-surface markers of murine macrophages and microglia [21,22]. This finding leads us to hypothesize that type I taste cells may play a pivotal role in the inflammatory process within papillae. Hence, we investigated the expression of other macrophage markers in GLAST-positive cells. The flow cytometry analysis of isolated TBC showed that type I TBC (GLAST-positive cells) represented more than 50% of the gustatory cells, in accordance with the literature [23]. GLAST-positive cells co-express F4/80 in 22.6%, CD11b in 3.24%, and CD64 in 1.87% of cases. We also observed that the mRNA expression of CD68, CD11c, F4/80, and TLR4 was higher in GLAST-positive cells compared to GLAST-negative cells. CD11b is a marker of macrophages and microglia. Moreover, CD11b allows adhesion and migration of microglia within the central nervous system and enhances their capacity to bind to target cells for phagocytosis. Glial cells and macrophages also express other markers, such as CD11c and CD64 [24]. CD11b is constitutively expressed, while CD11c is upregulated in active microglia [20]. Akinrinmade et al. [25] reported that CD64 is constitutively expressed by macrophages and its expression is highly increased following their activation.

Type II TBC express TLRs 2, 3, and 4; however, there is no report concerning the ex-pression of TLR by type 1 TBC. TLR4 is required for LPS responsiveness, as it recognizes a number of microbial-derived molecules and plays an essential role in the initiation of inflammation [26]. TLR4 is mainly found in macrophages and glial cells and is involved in host defense against gram-negative bacteria [27]. In addition, Feng et al. [28] demonstrated that TNFα production in taste bud cells is mediated through TLR pathway using TLR2^-/-^/TLR4^-/-^ double-knockout mice. These authors also reported that TNFα was produced by type II cells, but not by type I and type II cells. The present study is in contrast with the findings of Feng et al. [28] This could result from the cell manipulation and/or stimulation of type I cells under in vitro conditions.

Inflammation is a complex phenomenon which involves numerous factors, such as pro-inflammatory and anti-inflammatory cytokines. To evaluate if type I gustatory cells can differentiate in vitro to a phenotype similar to M1 or M2 macrophages, GLAST-positive cells were treated with IL-4, a cytokine known to be involved in the polarization of macrophages to M2 phenotype, or LPS, which is involved in the polarization of macrophages to the M1 phenotype [19,20]. Stimulation by IL-4 increased the mRNA expression of F4/80 and arginase 1 in GLAST-positive cells. Arginase 1 is mainly expressed in the IL-4-induced M2 phenotype and metabolizes arginine to ornithine and urea, limiting the conversion of L-arginine to NO by inducible nitric oxide synthetase (iNOS) [29]. We notice that the expressions of F4/80 and TNFα are inversely regulated, as reported by Gordon et al. [30]. It is well known that F4/80 is down-regulated in activated macrophages [31], whereas resident macrophages highly express F4/80 [32]. F4/80-positive macrophages have recently been shown to induce CD8+ regulatory T cell activity which suppresses antigen specific immunity [33]. Also, we report that LPS-treatment induces a M1-like macrophage phenotype characterized by increased mRNA expression of pro-inflammatory cytokines such as TNFα, IL-1β, and IL-6. Our observations corroborate the findings of Cohn et al. [15], who have observed that intraperitoneal injection of LPS induced the expression of several pro-inflammatory cytokines such as TNFα, IFN-γ, and IL-6, in mouse circumvallate and foliate papillae.

Obesity is associated with low-grade inflammation, characterized by the release of pro-inflammatory cytokines including TNFα, IL-1β, and IL-6 [34]. It has been reported that these pro-inflammatory cytokines are responsible for taste dysfunction by reducing taste bud cell abundance and inhibiting their renewal [35]. Growing evidence supports that obese people have a decreased taste perception that may lead to high calorie intake and, consequently, to obesity. Despite the fact that the presence of these pro-inflammatory molecules is observed in the lingual epithelium of obese animals [35], the cell type that releases pro-inflammatory cytokines remains to be clarified. After the isolation of type I GLAST-positive TBC from the lingual epithelium of obese mice, we observed that the mRNA expression of pro-inflammatory cytokines TNFα, IL-1β, and IL-6 was significantly higher than in lean animals. These results suggest that type I gustatory cells may also represent a source of pro-inflammatory cytokines responsible for the inflammation within taste papillae of obese mice. These results are confirmed by measuring cytokines in the supernatant of purified GLAST-positive cells. We observed that LPS, but not IL-4, induced TNFα and IL-6. Taken together, we demonstrated that type I gustatory cells may release pro-inflammatory cytokines during chronic inflammation in a pathological state like obesity. Hence, we conclude that type I gustatory cells may be involved in the regulation of the inflammatory process in both acute and chronic inflammation of taste papillae and, consequently, modulates taste sensitivity.

## 4. Materials and Methods

### 4.1. Animal and Diet

Animal experiments were conducted as per general guidelines for the care and use of laboratory animals, recommended by the council of European Economic Communities. The experimental protocol was also approved by the regional animal ethical committee of the University of Burgundy (Approval number 2014-17/12/2014-32).

Two months old male wild-type C57BL/6J mice were purchased from Janvier Labs (France). Mice were kept under constant laboratory conditions, i.e., temperature (25 °C) and humidity (60 ± 5%), with a 12-h light/dark cycle, and provided with either the following diets: standard laboratory diet (STD), purchased from SAFE (Augy, France), or high fat diet (HFD), prepared weekly as indicated in Table 1. 

### 4.2. Isolation of Taste Papillae

A mixture of elastase (2 mg/mL) and dispase (2 mg/mL; grade II), dissolved in Tyrode’s solution, was injected submucosally into three to four locations, around, and under the single circumvallate papilla of the tongue, which was then incubated in Ca^2+^-free Tyrode’s solution (140 mM, NaCl; 5 mM, KCl; 10 mM, HEPES; 2 mM EGTA; 10 mM, glucose; 10 mM, sodium pyruvate) for 10 min at 37 °C. The circumvallate papilla and lingual epithelium containing circumvallate papillae were extracted from the connective tissue with forceps and soaked in RPMI 1640 medium, containing 2 mM EDTA.

### 4.3. Culture of Mouse Taste Bud Cells (TBC)

The isolated circumvallate papillae were incubated in RPMI 1640 medium, containing 2 mM EDTA, 1.2 mg/mL elastase, 0.6 mg/mL collagenase (type I), and 0.2 mg/mL trypsin inhibitor at 37 °C for 20 min. Cells were separated from the digested tissue and transferred into a new tube, then centrifuged (600× *g* 10 min) and further resuspended in RPMI 1640 complete medium supplemented with 10% fetal bovine serum.

### 4.4. Magnetic Labeling and Isolation of GLAST Positives Cells

Labeling and separation of cells were performed using an anti-GLAST kit (ACSA-1) and a MicroBeads kit from Miltenyi Biotec (Paris, France), according to the manufacturer’s instructions. Briefly, up to 10 × 10^6^ taste bud cells (TBC) were suspended in 80 µL of PBS buffer, containing 0.5% BSA, added with FcR blocking reagent for 15 min at 4 °C, followed by incubation with 20 µL anti-GLAST antibodies conjugated to biotin for 10 min. After washing with 5 mL MACS buffer, TBC were incubated with 20 µL MicroBeads coupled to anti-Biotin anti-bodies for 15 min. TBC were then washed with MACS buffer and centrifuged (300× *g* 5 min) to remove any unbound beads from the pellet. The cell suspension was loaded onto an MS Column (Miltenyi Biotec), which was placed in a magnetic MACS Separator (Miltenyi Biotec). The column was washed three times with 500 µL of MACS buffer for elution of GLAST negatives cells, whereas the magnetically labeled GLAST-positive cells were retained within the column. The column was then removed from the GLAST separator, and the positive cells were eluted by rinsing the column with 1 mL of MACS buffer. Isolated gustatory cells were seeded overnight onto Petri dish wells before proceeding to further experiments.

### 4.5. Differentiation into M1 and M2 Macrophages-like Cells

In order to evaluate whether type I cells can differentiate either into the M1 or M2 macrophage-like phenotypes, glial positives cells were treated for 24 h with IL-4 (20 ng/mL) to induce a polarization toward M2 cells. LPS (100 ng/mL) was employed to induce a polarization toward M1 cells.

### 4.6. Immunocytochemistry of Type I Gustatory Cells

We performed immunocytochemical detection to confirm the co-expression of GLAST and F4/80 on purified GLAST positives cells. Hence, cells were seeded onto a Biocoat poly-D-lysine-coated coverslips in 24-well plate and cultured for 12 h. Coverslips were fixed in 95% ethanol and rehydrated in 0.1 M PBS, pH 7.4. Coverslips were blocked in PBS containing 5% fetal calf serum and 0.2% Triton X-100 for 30 min at room temperature, followed by overnight incubation with primary antibody anti-GLAST (20785-1-AP, Proteintech Group, Manchester, UK), at 1:200 dilution in PBS 2.5% FCS. After washing, coverslips were incubated for 45 min at room temperature with 1:500 anti-rabbit secondary antibody, coupled to Alexa Fluor-568. For double staining with F4/80 (ab300422, abcam, Paris, France), NTPDase2 (A31434, Antibodies.com), or Cytokeratin 8 (ab53280, abcam, Paris, France) antibodies, after washing, the slides were incubated again overnight with 1:200 primary antibodies (anti-F4/80, or anti-NTPDase2, or anti-Cytokeratin 8), followed by incubation with 1:500 secondary anti-rabbit antibodies, coupled to Alexa-488, as described for GLAST staining. After three washings with PBS, slides were incubated with DAPI to counterstain the nuclei. Finally, coverslips were removed and placed onto the slide in the presence of a drop of Aqua Poly-Mount mounting medium for analysis using a fluorescent microscope (Carl Zeiss Axioskop).

### 4.7. Flow Cytometry Analysis

The isolated TBC were washed with PBS by centrifugation (800× *g* 5 min, at 4 °C), then resuspended in 500 µL PBS containing 5% FCS (10^6^ cells/assay), and incubated for 30 min at 4 °C with antibodies (as cited above). After washing with 1 mL PBS 5% FCS, cells were analyzed using a FACSCalibur flow cytometer (Becton Dickinson, Franklin Lakes, NJ, USA) and data were processed using the FlowJo software, version 10 (Tree Star, San Carlos, CA, USA). Cell surface molecular expression was monitored using the following fluorochrome-labeled antibodies: anti-GLAST PE (ref 130-118-344) and anti-CD11b FITC (ref 130-113-234) diluted at 1:50; anti-CD64 APC diluted at 1:20 and anti-F4/80 PE Vio 615 (ref 130-107-709) diluted at 1:10. All antibodies were from Miltenyi Biotec (Paris, France).

### 4.8. Determination of mRNA Expression

Real-Time PCR (RT-PCR) was performed to evaluate mRNA expression of GLAST, CD68, CD11c, TLR4, F4/80, Arginase 1, TNFα, IL-4, IL-6, and IL-1β. Total RNA was isolated using TRIzol Reagent (Invitrogen, Waltham, MA, USA), treated with DNase using the RNase-free DNase Set (Life Technologies, Carlsbad, CA, USA), reverse transcribed using M-MLV Reverse Transcriptase (Invitrogen, Waltham, MA, USA), and examined by RT-qPCR using predesigned primers and probes. Amplification was performed using SYBR^®^ Green PCR Master Mix (Life Technologies, Carlsbad, CA, USA) as described [36]. Primers for the genes of interest were designed using the Beacon Designer software. β-Actin was used as endogenous control. The relative quantity of mRNA was calculated with the 2^(−∆∆CT)^ method [37]. The primers used are shown in Table 2.

### 4.9. Determination of Pro-Inflammatory Cytokines Levels

TNFα and IL-6 were measured in the supernatants of type I cells treated for 24 h with IL-4 or LPS using ELISA kits (BioLegend, San Diego, CA, USA) according to the manufacturer’s guidelines.

### 4.10. Statistical Analyses

Data are presented as mean ± SEM. The statistical difference between groups was evaluated by Student’s *t*-test or one-way ANOVA test followed by Dunnett T3 test, based on the outcome of a Shapiro–Wilk test. All statistical analyses were conducted using GraphPad Prism version 6.01 (GraphPad Software, Boston, MA, USA). *p* values < 0.05 were considered statistically significant.

## Figures and Tables

**Figure 1 ijms-24-10325-f001:**
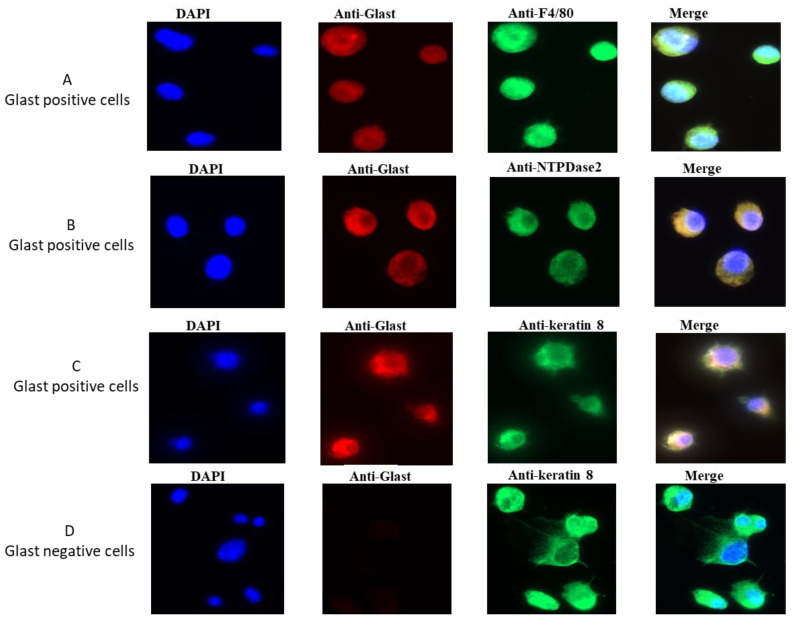
GLAST-positive gustatory cells express F4/80. GLAST-positive gustatory cells were stained with anti-GLAST and anti-F4/80 (**A**), anti-NTPDase2 (**B**), or anti-keratin 8 (**C**) antibodies. GLAST-negatives gustatory cells were stained with anti-GLAST and anti-keratin 8 antibodies (**D**). Nuclei were counterstained with DAPI, and slides were visualized using immunofluorescence microscopy (40×). Merged images of the red, green, and blue fluorescence are shown (Merge).

**Figure 2 ijms-24-10325-f002:**
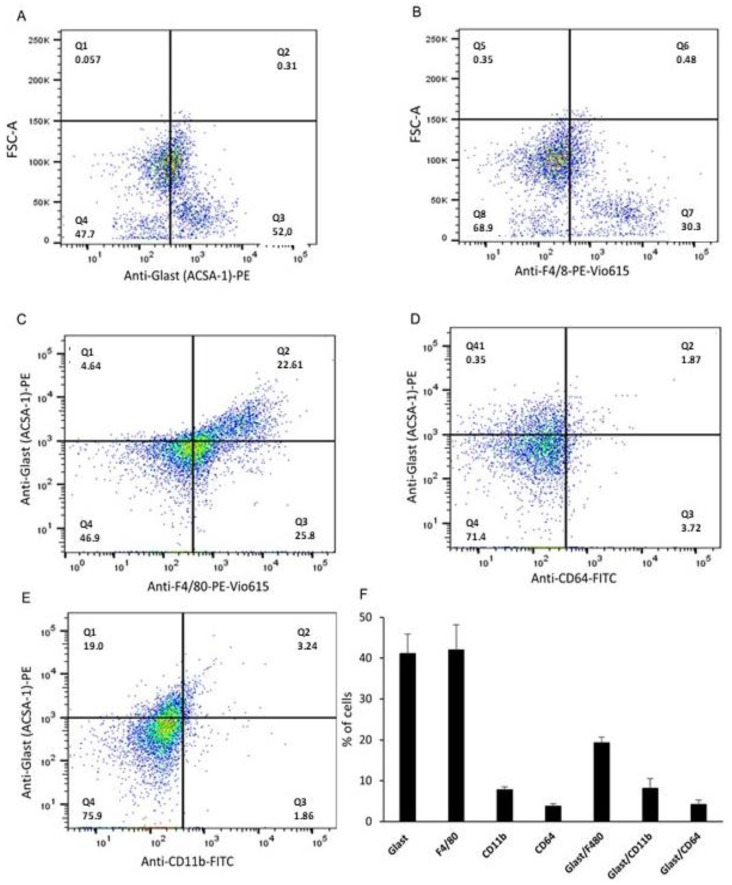
Expression of different inflammation markers by GLAST-positive gustatory cells using flow cytometry. Cell surface molecular expression was monitored using the following fluorochrome-labeled antibodies: anti-GLAST PE, anti-F4/80 PE Vio 615, anti-CD64 APC, and anti-CD11b FITC. (**A**) represents GLAST-positive gustatory cells. (**B**) represents a subpopulation of cells expressing F4/80. (**C**) represents co-expression of GLAST and F4/80, (**D**) represents co-expression of GLAST and CD64, while (**E**) represents co-expression of GLAST and CD11b. (**F**) The histograms represent the percentage of cells expressing different markers (mean ± SEM, *n* = 3).

**Figure 3 ijms-24-10325-f003:**
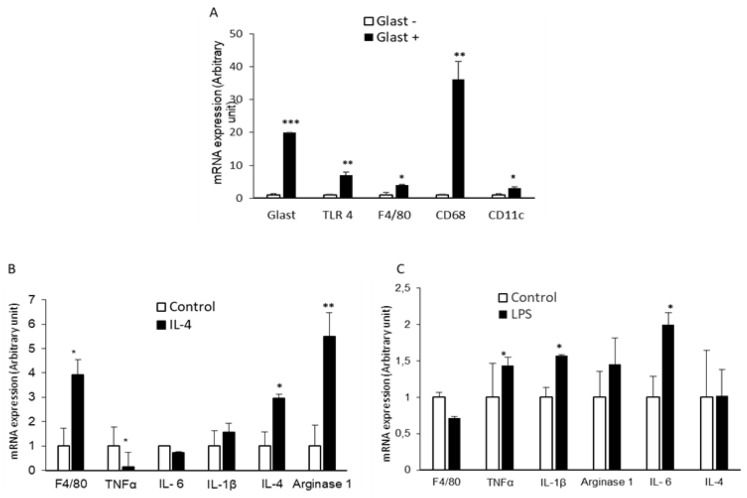
mRNA expression of different markers in GLAST-positive cells. qRT-PCR was per-formed to quantify mRNA expression of GLAST, TLR4, F4/80, CD68, and CD11c, in GLAST-positive and GLAST-negative cells (**A**). The GLAST-positive cells express mRNA encoding F4/80, TNFα, IL-6, IL-1β, IL-4, and arginase 1 (**B**,**C**). Cells were activated by IL-4 at 20 ng/mL (**B**) or LPS at 100 ng/mL (**C**), as explained in “Materials and Methods”. Data are presented as means ± SEM (*n* = 3). The symbols *, **, and *** represent *p* < 0.05, *p* < 0.01, and *p* < 0.001, respectively, compared to the corresponding control. *p* values were obtained using Student’s *t*-test.

**Figure 4 ijms-24-10325-f004:**
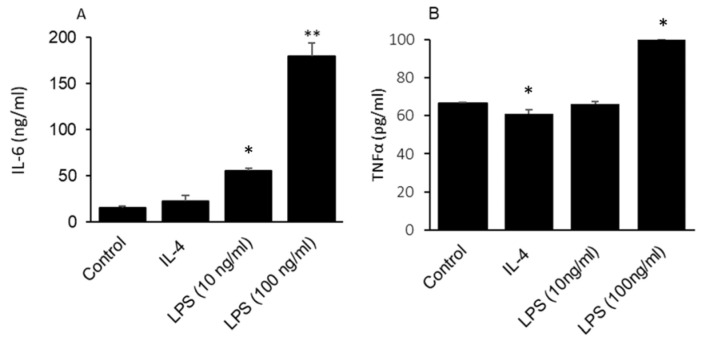
Concentrations of pro-inflammatory cytokines in cultured GLAST-positive gustatory cells. Cultured GLAST-positive cells were treated with 20 ng/mL of IL-4 or 100 ng/mL LPS. IL-6 (**A**) and TNFα (**B**) concentrations in cultured supernatants of GLAST-positive cells were determined by ELISA. Histograms represent the results of independent experiments (*n* = 3). The symbols * and ** represent *p* < 0.05 and *p* < 0.01, respectively, compared to control. *p* values were obtained using one-way ANOVA tests followed by Dunnett T3 tests.

**Figure 5 ijms-24-10325-f005:**
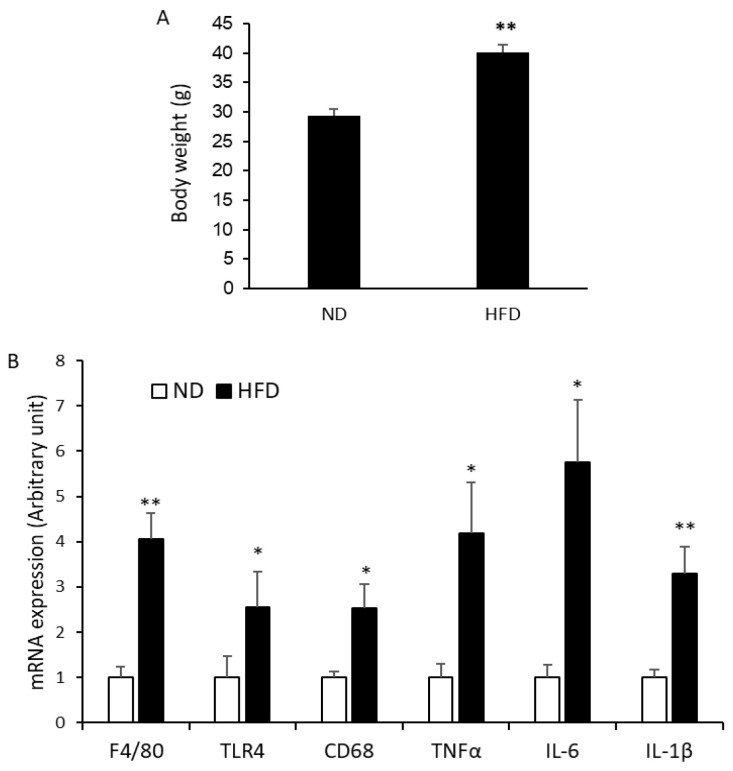
The effect of HFD on body weight and inflammatory markers in type 1 taste gustatory cells. Mice were maintained on STD or HFD. After 12 weeks of experimentation, animals were weighed (**A**) before sacrifice by cervical dislocation, and tongues were removed to isolate type 1 gustatory cells to assess expression of mRNA encoding F4/80, TLR4, CD68, TNFα, IL-6, and IL-1β (**B**). Data are presented as means ± SEM (*n* = 10). The symbols * and ** represent *p* < 0.05 and *p* < 0.01, respectively, as compared to ND-fed animals. *p* values were obtained by using Student’s *t*-test.

**Table 1 ijms-24-10325-t001:** Diet composition.

Diet Composition of the Scheme g/100g A03 SAFE	Normal Diet		High-Fat Diet (60%)	
	g	kcal	g	kcal
Carbohydrate	52.00	208	35.43	141.73
Protein	21.40	85.6	14.58	58.33
Lipids	5.10	45.9	35.34	318.02
Fibers	4	8	2,73	5.45
Minerals Ash	5.40	0	3.68	0
water	12.10	0	8.24	0

Animals had free access to food and water. After 12 weeks of experimentation, animals (STD, *n* = 10 and HFD, *n* = 10) were sacrificed by cervical dislocation, tongues were removed immediately, and put into a Petri dish containing Tyrode’s solution (120 mM, NaCl; 5 mM, KCl; 10 mM, HEPES; 1 mM, CaCl_2_; 1 mM, MgCl_2_; 10 mM, glucose; 10 mM, sodium pyruvate).

**Table 2 ijms-24-10325-t002:** Sequence of the primers used in the study.

Gene	Primer Sequence	Accession Number
GLAST	F: 5′ ACCAAAAGCAACGGAGAAGAG 3′ R: 5′ GGCATTCCGAAACAGGTAACTC 3′	NM_148938.3
CD68	F: 5′ CTTCCCACAGGCAGCACAG 3′ R: 5′ AATGATGAGAGGCAGCAAGAGG 3′	NM_001291058.1
CD11c	F: 5′ GTGCCCATCAGTTCCTTACA 3′ R: 5′ GAGAAGAACTGTGGAGCTGAC 3′	NM_021334.3
TLR4	F: 5′ AGCTCCTGACCTTGGTCTTG 3′ R: 5′ CGCAGGGGAACTCAATGAGG 3′	NM_025817.4
F4/80	F: 5′ TGACTCACCTTGTGGTCCTAA 3′ R: 5′ CTTCCCAGAATCCAGTCTTTCC 3′	NM_001355722.1
Arginase 1	F: 5′ CTCCAAGCCAAAGTCCTTAGAG 3′ R: 5′ AGGAGCTGTCATTAGGGACATC 3′	NM_007482.3
TNFα	F: 5′ CTGTTCTCATTCCTGCTTGTGG 3′ R: 5′ AATCGGCTGACGGTGTGG 3′	NM_001278601.1
IL-4	F: 5′ CGAGCTCACTCTCTGTGGTG 3′ R: 5′ TGAACGAGGTCACAGGAGAA 3′	NM_021283.2
IL-6	F: 5′ TAGTCCTTCCTACCCCAATTTCC 3′ R: 5′ TTGGTCCTTAGCCACTCCTTC 3′	NM_001314054.1
IL-1β	F: 5′ CACAGCAGCACATCAACAAG 3′ R: 5′ GTGCTCATGTCCTCATCCTG 3′	NM_008361.4
β-Actin	F: 5′ CTGTGTTGTCCCTGTATGCG 3′ R: 5′ TAGATGGGCACAGTGTGGGT 3′	NM_007393.5

## Data Availability

The data presented in this study are available on request from the corresponding author.

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
