# Peer review of "In Vitro Functional Characterization of Type-I Taste Bud Cells as Monocytes/Macrophages-like Which Secrete Proinflammatory Cytokines"

_ijms, 2023, doi:10.3390/ijms241210325_

Round 1

Reviewer 1 Report

The authors report that isolated Type I, glial-like cells in the taste bud express the classical tissue macrophage marker, F4/80.  Cultured cells expressing the glial marker, GLAST, were treated to elicit M1 or M2 cytokine responses in vitro.  Proinflammatory cytokine expression increased in GLAST+ cells in male mice fed a high-fat diet for 12 weeks compared to control diet fed mice.  The authors conclude that Type I taste cells have a macrophage-like role in taste cells, and become polarized to a pro-inflammatory phenotype in obesity.  A number of methodological details are missing and there appears to be a problem with the statistical test chosen in some experiments.  The major concern, however, is whether this GLAST+ population actually represents Type I taste cells in vivo. 

Introduction, first sentence.  Rewrite to reflect that papillae are taste bud-containing structures in the lingual epithelium, rather than …”which are located in the lingual epithelium, called papillae”.

Results

·       Please introduce GLAST, species used, and other basic methodological details need to understand the results.

·       How was purity assessed?

·       In the methods, these cells are referred to as “freshly” isolated but these are completely rounded cells which have lost cell polarity.

·       It would be more convincing to first show that the GLAST+ cells also express Type I markers like NTPdase2 and a general taste cell marker like keratin 8.  Positive (i.e. macrophages) and negative controls are also needed.

·       Importantly, is F4/80 expressed in taste cells in vivo?  A number of studies show staining for F4/80, CD68 or other macrophage markers in lingual tissue sections and taste cells are negative.  This implies that upregulation of F4/80 here, ex vivo, reflects culture conditions, cell contamination, or other technical reason. 

·       Figure 2 caption needs basic details about labeling for flow cytometry to understand the figure.

·       Figure 3C legend.  IL-4 is a typical M2 stimulus, but I’m confused why both LPS and an anti-IL-4 (antibody?) were added to induce an M1 phenotype.

·       Figure 4.  Student’s t test is not appropriate to make multiple comparisons unless a correction factor is applied.

Discussion

Third sentence.  The loss or recovery of taste function from COVID-19 has not been shown to depend on IL-6.  The cited paper from a small patient sample shows a barely significant correlation between circulating IL-6 and taste perception, not causation or mechanism.

Methods

·       The high-fat diet is not described, as Table 1 has primer information not diet.

·       Antibody sources for ICC are missing.

·       Concentration of IL-4 and LPS and cytokine source are missing.

Some editing is needed

Author Response

The reviewer pointed out that there were a number of methodological details that were missing and we should look at the the statistical test chosen in some experiments.  We would like to state that we have re-read and rewritten the methodological details in the revised version of the MS. We have employed some new statistical tests that are mentioned appropriately in the text.

Main points raised:

-We have rephrased the first sentence of the Introduction section as per suggestion of the referee.

 -In the Results section, we have introduced GLAST species as we have indicated that the purified cells were GLAST1+ cells as we used ACSA-1 antibody in our experiments.

-As regards the question on the purity of the GLAST+ cells. We would like to state that the percentage of GLAST1+ cells purified by magnetic cell separation should be around 95% as reported by the manufacturer (Miltenyi Biotec) and by Jungblut M et al. (2012). Moreover, the figure 1 (immunofluorescence) shows that all the cells purified with the anti-ACSA-1 antibody were GLAST+ cells.

(Jungblut M, Tiveron MC, Barral S, Abrahamsen B, Knöbel S, Pennartz S, Schmitz J, Perraut M, Pfrieger FW, Stoffel W, Cremer H, Bosio A. Isolation and characterization of living primary astroglial cells using the new GLAST-specific monoclonal antibody ACSA-1. Glia. 2012 May;60(6):894-907)

-The referee mentioned that in the Methods, these cells were referred to as “freshly” isolated but these are completely rounded cells which have lost cell polarity. Hence, we have removed the word “freshly” from the sentence as after enzymatic tissue digestion, cell could lose their polarity.

-The referee further raised the question that it would be more convincing to first show that the GLAST+ cells also expressed Type I markers like NTPdase2 and a general taste cell marker like keratin 8.  Positive (i.e. macrophages) and negative controls were also needed.

We have now added an “Insert” in the revised M/S in Fig 1 IF, with double staining : GLAST-positive gustatory cells were stained with anti-GLAST and anti-F4/80 (A) or anti-NTPDase2 (B) or Anti-Cytokeratin 8 (C) antibodies, GLAST-negative gustatory cells were stained with anti-GLAST and Anti-Cytokeratin 8.

As negative control macrophages,  RAW cells  were used with a double staining with anti-GLAST and anti-F4/80. We can observe that Raw cells which were  F4/80 positive were indeed GLAST negative. We did not include this result in the MS as it does not bring new information in the MS (see below).

- The referee mentioned that a number of studies showed with staining for F4/80, CD68 or other macrophage markers in lingual tissue sections that taste cells were negative.  It would be possible that upregulation of F4/80 here, ex vivo, reflected culture conditions (or cell contamination, or other technical reason ?). 

Hence, we would like to say that despite that Type I cells represent the half of mammalian taste bud cells, few details are known regarding their function. Type I cells has been proposed to regulate the ionic environment in taste buds (Dvoryanchikov G, 2009). Type I cells are termed as “glial-like” because they express glial glutamate transporter (GLAST) and the ectonucleotidase, NTPase2 (Bartel et al., 2006) that breaks down ATP released by Type II and thus plays play a function in clearing neurotransmitters like glial cells. In addition, Type I cells like glial cells,  release GAB and express GAD65 (Dvoryanchikov G. 2011).  Hence, it is not impossible that Glial cells do not express F4/80 as we observed in the FACS results.

Reference :

- Dvoryanchikov G, Sinclair MS, Perea-Martinez I, Wang T, Chaudhari N (2009) Inward rectifier channel, ROMK, is localized to the apical tips of glial-like cells in mouse taste buds. J Comp Neurol 517:1–14. 10.1002/cne.22152

- Bartel DL, Sullivan SL, Lavoie EG, Sevigny J, Finger TE (2006) Nucleoside triphosphate diphosphohydrolase-2 is the ecto-ATPase of Type I cells in taste buds. J Comp Neurol 497:1–12. 10.1002/cne.20954

- Dvoryanchikov G, Huang YA, Barro-Soria R, Chaudhari N, Roper SD. GABA, its receptors, and GABAergic inhibition in mouse taste buds. J Neurosci. 2011; 31(15):5782-91.

-In the Figure 2, we have added more details on the captions as was asked by the reviewer.

-The referee questioned regarding the Figure 3C why we used both LPS and an anti-IL-4 (antibody) to induce an M1 phenotype. We did not use anti-IL-4 antibody and it was, in fig 3, a typing mistake. We regret for it. We also thank the reviewer for this comment.

-The referee pointed out that in Figure 4, Student’s t test is not appropriate to make multiple comparisons unless a correction factor is applied. Now, we have used, in revised MS, ANOVA One-Way test followed by Dunnett T3 test.

-We have now mentioned the composition of the high-fat diet in the revised MS (see, Table 1). We have also added the antibody sources and information on the concentration of IL-4 and LPS.

Reviewer 2 Report

(TBC), namely Type I, Type II and Type III. Type I TBC, looking at GLAST (glutamate-aspartate transporter), is a thesis with various inverse terms that reverse the role of cells called glial-like cells and disciples. Type I taste cells share many rights with macrophages and contributed to providing evidence that they can contribute.

Author Response

The referee did not raise any specific question. We have reread the MS and brought some more calcifications.  

Reviewer 3 Report

Peer Review for IJMS: MDPI

Hichami et al:  Functional Characterization of Type-1 Taste Bud Cells……

Over All Comments:  Wonderfully written with a very few very minor suggestions for typos, etc.  Introduction provided background for study, methods excellently written, results were convincing (good figures), and discussion clearly supported/explained the findings.  This was as clean of a paper as I have peer-reviewed in quite some time….well done! I cannot wait to see this in print I the near future and yes I also agree that our sense of taste may well influence our risk of developing obesity.

Title:  Ok as is, but authors “may” wish to look at term “monocyte-macrophages”  maybe choose just word monocyte or macrophage or put this on one line (one-hyphen, not two)??

Abstract: Excellent writing, no edits needed

Introduction: Excellent writing, one a couple suggestions

L31….components on the tongue….

L58:  Most of your writing in manuscript uses “paste tense”, this would look nicer like this I think:  The present study was designed…..

Methods: Excellent writing

L267: …performed using an Anti-GLAST…..

Results: Excellent writing

L66: Type 1 cells purified using the Anti-GLAST (ACSA-1) and…..

L68: Double labelling using….

L111-113:  Suggest rewording as written: “macrophages, we wanted to determine

if….   “tempted” might mean you did not bother to do this…..

L130: Fig4  Delete thus part following the manufacturers instructions….self explanatory to me at least.

L139: Fig5. Delete as explained in the “Materials and Methods” section.

Discussion: Good handling of your findings and what it means.  I agree with you on this one, nice work!

L154:  Might reword this one of two ways……evidence indicates that the taste system….  OR  evidence indicates that taste systems might….

L171:L …current study was to determine if type 1 TBC…..

L186: “reported”…not re-ported  (simple typo)

Author Response

 The reviewer appreciated our study. However he raised minor concerns : We have added just one hyphen in the Title of the MS. We have reworded some sentences as per advice of the reviewer.

Round 2

Reviewer 1 Report

The authors have adequately addressed most of my comments.  However, they did not demonstrate that type I taste cells express macrophage markers in vivo, in contrast to several published studies.  They show that type I taste cells can be induced to express macrophage markers in vitro.  At the very least, the authors should discuss how their findings oppose published work in which these markers are not expressed in type I taste cells in tissue sections, cite those studies, and add that these findings may not be relevant in vivo.  If they cannot confirm the expression of macrophage markers in vivo, then that should be acknowledged and the qualifier "in vitro" added to the title.

Adequate but typos (e.g. NTPdase2)

Author Response

The main concern of the reviewer was that our findings were based on in vitro experiments and hence, we should cliam what we observed was under such conditions. We have changed the title of the MS and it starts with "In vitro ...".

Besides, the referee advised us to cite the paper on the fcat that type I cells do not release TNFa. We have cited the reference of Feng and discussed thier findings in the Discussion section. We have re-read the MS and corrected the minor errors. We thank the referee for the pertaining remarks. We hope that the MS will find a suitable place in the Journal.

Round 3

Reviewer 1 Report

Acceptable

Acceptable